# Genetic Manipulation of the Brassicaceae Smut Fungus *Thecaphora thlaspeos*

**DOI:** 10.3390/jof7010038

**Published:** 2021-01-09

**Authors:** Lesley Plücker, Kristin Bösch, Lea Geißl, Philipp Hoffmann, Vera Göhre

**Affiliations:** Institute of Microbiology, Cluster of Excellence in Plant Sciences, Heinrich-Heine University, Building 26.24.01, Universitätsstr.1, 40205 Düsseldorf, Germany; lesley.pluecker@hhu.de (L.P.); kristin.boesch@hhu.de (K.B.); lea.geissl@hhu.de (L.G.); Philipp.Hoffmann.15@uni-duesseldorf.de (P.H.)

**Keywords:** mating, pheromone receptor, homologous recombination, infection, smut, transformation, protoplast

## Abstract

Investigation of plant–microbe interactions greatly benefit from genetically tractable partners to address, molecularly, the virulence and defense mechanisms. The smut fungus *Ustilago maydis* is a model pathogen in that sense: efficient homologous recombination and a small genome allow targeted modification. On the host side, maize is limiting with regard to rapid genetic alterations. By contrast, the model plant *Arabidopsis thaliana* is an excellent model with a vast amount of information and techniques as well as genetic resources. Here, we present a transformation protocol for the Brassicaceae smut fungus *Thecaphora thlaspeos*. Using the well-established methodology of protoplast transformation, we generated the first reporter strains expressing fluorescent proteins to follow mating. As a proof-of-principle for homologous recombination, we deleted the pheromone receptor *pra1*. As expected, this mutant cannot mate. Further analysis will contribute to our understanding of the role of mating for infection biology in this novel model fungus. From now on, the genetic manipulation of *T. thlaspeos*, which is able to colonize the model plant *A. thaliana,* provides us with a pathosystem in which both partners are genetically amenable to study smut infection biology.

## 1. Introduction

Smut fungi are important pathogens causing economic losses in crops such as barley, wheat, maize, and potato [1]. The dimorphic lifecycle of grass smut fungi is comprised of a yeast form [2] that, in contrast to many other biotrophic fungi, can be cultured, and is amenable to genetic manipulation [3]. In combination with the efficient homologous recombination, this has turned *Ustilago maydis*, the maize smut fungus [4], into an important model organism for fungal and infection biology [5,6].

*U. maydis* starts the infection by mating, resulting in the morphological switch to the infectious filamentous form. Mating in smut fungi is controlled by two mating loci. The *a* locus encodes for pheromones (*mfa*) and pheromone receptors (*pra*) that mediate recognition of compatible mating partners and trigger cell fusion [7]. The *b* locus contains two genes (*bW*, *bE*) encoding for subunits of a heterodimeric, homeodomain transcription factor. When an active transcription factor is assembled from different alleles in the common cytosol after plasmogamy, filamentous growth is initiated and thereby the fungus switches from saprophytic to pathogenic growth [8]. Notably, the infectious filaments are arrested in the cell cycle until successful penetration of the host plant [9]. This mating system is widely conserved in grass smut fungi and allows for genetic exchange between the mating partners [10].

In contrast to the well-characterized grass smut fungi infecting important monocot crop plants with *U. maydis* at the forefront, little is known about smut fungi infecting dicot plants. A reemerging example is *Microbotryum* that is regaining attention today [11]. By contrast, the *Thecaphora*-clade [12], with agronomically relevant members such as *T. solani* [13] or *T. frezii* [14] infecting potato and peanut, respectively, remains largely elusive. One member, the Brassicaceae smut fungus *T. thlaspeos*, infects several Arabis species [12] and can colonize *Arabidopsis thaliana* [15], making it a good system to study smut infection in model plants. Interestingly, germinating teliospores of *T. thlaspeos* directly give rise to an infectious filament, and no saprophytic phase is known. However, prolonged cultivation leads to fungal proliferation as filamentous, haploid cultures. From such cultures, two filamentous haploid *T. thlaspeos* strains of compatible mating types, LF1 and LF2, could be isolated [15]. The unique germination pattern and the emergence of haploid filaments in culture raise questions about mating and meiosis in *T. thlaspeos*.

The genome of *T. thlaspeos* was recently sequenced and annotated. With 20.5 Mb, 6239 gene models, and a low repeat content, it is a typical smut fungal genome [16]. Notably, we could identify the mating loci *a* and *b* in this genome. In comparison to the grass smut fungi, the *a* locus of *T. thlaspeos* is strongly rearranged, and aligns well to the biocontrol yeast *Anthracocystis flocculosa* (formerly *Pseudozyma flocculosa*). It still contains one copy of the pheromone receptor *pra1* and pheromone *mfa1*, or *pra2* and *mfa2* in *T. thlaspeos* LF1 and LF2, respectively [16]. The *b* locus is conserved with a bi-directional promoter in between the two genes, and we have previously shown that the heterodimer formation of bE and bW isconserved in *T. thlaspeos* [15]. Conservation of the mating genes throughout evolution suggests that *T. thlaspeos* still uses mating, e.g., for exchange of genetic material. However, the infectious lifecycle so far has not revealed a stage where mating is required. Therefore, we aimed for a more detailed understanding of the role of the *T. thlaspeos* mating genes.

To study the mating process in *T. thlaspeos*, genetic manipulation is essential. For example, reporter lines enable life-cell imaging, and deletion mutants can give insight into mutant phenotypes. Several transformation methods have been developed for smut fungi. Protoplast-mediated transformation is used for the grass smuts *U. maydis* [17], *Sporisorium reilianum* [18], *U. hordei* [19], *U. esculenta* [20], and *U. bromivora* [21], as well as the filamentous Basidiomycete *Serendipita indica* (formerly *Piriformospora indica*) [22], and the biocontrol yeast *An*. *flocculosa* [23]. Agrobacterium-mediated transformation is used for *S. scitamineum* in combination with a CRISPR-Cas9 system [24], and also for bringing larger fragments into *U. hordei* [25] and for gene tagging in *U. maydis* [26].

Here, we have established a protoplast-based PEG-mediated transformation system and generated targeted deletion mutants to investigate the mating process in *T. thlaspeos*.

## 2. Materials and Methods

### 2.1. Fungal Culture Conditions

*T. thlaspeos* LF1 and LF2 haploid strains [15] from our own collection were used in this study. Both were grown in YEPS light liquid medium (1% yeast extract *w*/*v*, 0.4% *w*/*v* Bacto TM-Peptone, and 0.4% *w*/*v* sucrose) or YEPS light solid medium with 0.6% *w*/*v* plant agar or 1% *w*/*v* phytagel at 18 °C. Cryostocks of *T. thlaspeos* were generated by mixing exponentially growing cultures with 30% glycerol in growth medium followed by immediate freezing at −80 °C. Filamentous cultures were started by resuspending the glycerol stock in growth medium or plating the cells on solid medium as described above.

### 2.2. Plasmid Construction

All plasmids used in this study were generated with the Golden Gate Cloning technique (Appendix A) as described [27]. The *hpt-egfp* and *hpt-mcherry* sequences were codon-optimized for *U. maydis*. Promoter and terminator sequences from *T. thlaspeos* or *U. maydis* were amplified via PCR from genomic DNA.

### 2.3. Protoplasting

*T. thlaspeos* cultures were grown in YEPS light medium to an OD_600_ of 0.5–0.8 for 3 to 4 days. Fungal mycelium was collected using cell strainers with 40 µm mesh size (VWR TM Darmstadt, Germany) and washed with protoplasting buffer (0.1 M sodium citrate, 0.01 M EDTA 1.2 M MgSO_4_, and pH 5.8) to remove residual culture medium. *T. thlaspeos* tissue was resuspended in 9 mL protoplasting buffer, supplemented with 10 mg/mL Yatalase (Takara Bio, Kusatsu, Japan) and 20 mg/mL Glucanex (Sigma-Aldrich, St. Luis, MI, USA) per 100 mL cell culture, and incubated for 30–60 min at room temperature. Protoplast formation was controlled microscopically. When protoplasting was finished, protoplasting buffer was added to a total volume of 24 mL. Aliquots of 6 mL crude protoplast solution were overlayed with 5 mL trapping buffer (0.6 M sorbitol, 0.1 M Tris/HCl pH 7.0) and centrifuged at 4863× *g* (5000 rpm) in a swing out rotor at 4 °C for 15 min. The interphase was collected from all tubes and diluted with 2 volumes of ice-cold STC buffer (0.01 M Tris/HCl pH 7.5, 0.1 M CaCl_2_, and 1.0 M sorbitol). Protoplasts were pelleted at 4863× *g* (5000 rpm) in a swing out rotor at 4 °C for 10 min and resuspended in 500 µL ice-cold STC buffer. 100 µL protoplast aliquots were used for transformation immediately. A bullet point version of the protocol is available with the Appendix A.

### 2.4. Transformation

Transformation of *T. thlaspeos* protoplasts were carried out as described for *U. maydis* [28] with slight modifications. Transformation reactions were spread on YMPG-Reg (0.3% *w*/*v* Yeast extract, 0.3% *w*/*v* malt extract, 0.5% *w*/*v* Bacto-Peptone, 1% *w*/*v* glucose, 1 M sucrose, 0.6% *w/v* plant agar, Duchefa Biochemie, Haarlem, Netherlands) medium and incubated at 18 °C until colonies appeared (1–2 months). Selection was carried out on 10 µg/mL Hygromycin B (Roth, Karlsruhe Germany) following the layered-plate strategy used for *U. maydis* [28]. In this setup, hygromycin was provided in the bottom layer so that it took time to diffuse to the top before the selection took place. This gave protoplasts time to regenerate and express the resistance gene [28]. Colonies were singled-out on YEPS light solid medium supplemented with 10 µg/mL hygromycin. Single colonies were then used to inoculate YEPS light liquid cultures for molecular analysis.

### 2.5. Molecular Analysis of Transformants

Successful integration of the constructs were determined by PCR and Southern Blot analysis [28], and eGfp or mCherry fluorescence was used as a rapid indicator for expression of the constructs. Genomic DNA of *T. thlaspeos* was extracted using the NEB Monarch Genomic DNA Purification Kit (New England Biolabs, Frankfurt, Germany).

### 2.6. Mating Assay

For confrontation assays, liquid cultures of *T. thlaspeos* strains were spotted on YEPS light solid medium in close proximity and allowed to grow towards each other. When the hyphae of both strains were close enough to appear in the display window of the microscope, a time lapse experiment was conducted to monitor mating for 24–72 h.

For liquid mating assays, strains of compatible mating types were mixed in YEPS light liquid medium in equal amounts and incubated at 18 °C and 200 rpm. Medium was exchanged twice a week. After 8–14 days, plasmogamy was observed microscopically via eGfp and mCherry fluorescence.

### 2.7. Microscopy

Fluorescence microscopy as well time lapse experiments were performed on a Zeiss Axio Immager M1 according to [29]. Microscope control, image acquisition, and processing were done with the software package Meta-Morph (version 7; Molecular Devices).

## 3. Results

### 3.1. Protoplast—Generation from Filamentous T. thlaspeos Cultures

Protoplast-mediated transformation is well-established for fungi [30]. The protocol for *U. maydis* [17] was successfully adapted for other smut fungi and therefore was also the starting point for *T. thlaspeos* protoplast generation. Critical factors besides cultivation conditions are the enzyme mixture, the buffer composition, and the osmotic stabilizer. First, we compared different enzyme mixtures. Glucanex, a mix of lysing enzymes from *Trichoderma harzianum* including beta-1,3-glucanase activity, works well for *U. maydis* and the filamentous Basidiomycete *S. indica* [22]. Yatalase comprises of a mix of lysing enzymes from Corynebacterium, including chitinase-, chitobiase-, and beta-1,3-glucanase activity, for cell wall lysis of filamentous fungi. In combination with Glucanex, it is used for *U. bromivora* [21] and *Agrocybe aegerita* [31], or supplemented with chitinase for *Aspergillus niger* [32]. Novozyme 234, which worked very well for *U. maydis* [28], is no longer commercially available, so we did not include it in our study. In pilot studies, we compared enzyme and buffer combinations of published protoplasting protocols and found that the combination of Glucanex and Yatalase efficiently protoplasts the *T. thlaspeos* filaments (Appendix A).

Therefore, to first optimize the osmotic stabilizer, we used this enzyme mix in the *U. maydis* protoplasting buffer. Typical osmotic stabilizers are inorganic salts, sugars, or sugar alcohols [33]. For example, sorbitol is used for *U. maydis*, and sucrose for *U. esculenta*. Thus, we tested sorbitol and sucrose, as well as MgSO_4_, which is used frequently in combination with Yatalase. Most protoplasts were obtained using MgSO_4_ (Figure 1). Notably, sorbitol and sucrose inhibited cell wall lysis, confirming early observations [34,35]. Subsequently, we tested various commonly used protoplasting buffers, in combination with MgSO_4_ as the osmotic stabilizer. There were no significant differences between the four tested buffers (Table 1), but we observed tendencies that citrate buffers work better for fast growing cultures that were sub-cultured bi-weekly (Appendix A). Ultimately, we decided on the 0.1 M citrate buffer and 0.01 M EDTA, which is also the buffer used for *An. flocculosa*, the closest homolog of *T. thlaspeos,* and a bi-weekly splitting rhythm.

An advantage of MgSO_4_ as osmotic stabilizer is that the majority of intact protoplasts in the presence of MgSO_4_ have large vacuoles [38], which enables collection and purification, and floating in a trapping buffer [36]. Intact protoplasts accumulate in a sharp band at the interphase, and debris pellet at the bottom (Figure 2). After washing with STC buffer, up to 10^8^ protoplasts/g fresh weight can be recovered.

Upon determination of the optimal buffer and osmotic stabilizer, we reevaluated the composition and concentration of the enzyme mix (Figure 3). As the pilot study had indicated, for efficient degradation of the *T. thlaspeos* cell wall, the combined activity of Yatalse and Glucanex is necessary. As expected, individually the enzymes are poorly active, leaving filaments behind (Figure 3A). This emphasizes the importance of testing various lysing enzymes, alone and in combination, in different buffers to find a mix suitable for the organism of choice and its individual cell wall composition [30].

Finally, we aimed to decrease the enzyme concentrations to save costs. However, lowering the concentration to half resulted in incomplete digestion of the fungal cell wall after 30 min (Figure 3B). Since it was described earlier that a shorter incubation time is preferable, compared to a low enzyme concentration, regarding protoplast viability [39,40], we did not reduce the enzyme concentrations.

In contrast to other applications of protoplasts, for genetic manipulation, the protoplasts have to be viable and able to regenerate their cell wall. Thus, we next investigated the influence of the osmotic stabilizer on regeneration of the protoplasts after transformation. This allowed us to separate optimization of both steps. Using protoplasts generated in the presence of MgSO_4_, we assessed regeneration media containing different osmotic stabilizers such as sucrose, glucose, sorbitol, and KCl (Figure 4). MgSO_4_ was excluded due to incompatibility with the gelling agent. Maximal regeneration was obtained with 1 M sucrose as osmotic stabilizer, followed by sorbitol and glucose, while 1 M KCl completely inhibited regeneration and fungal growth (Figure 4A). In support, *T. thlaspeos* LF1 cell cultures do not grow on 1 M KCl, indicating that it is toxic at this concentration, while sucrose, glucose, and sorbitol only reduce the growth rate (Figure 4B). Without an osmotic stabilizer, cells were not able to regenerate, indicating that any residual filaments were efficiently removed during purification of the protoplasts (Figure 4A). *U. maydis* protoplasts regenerate into yeast cells and form colonies within three days [17,41]. For *T. thlaspeos*, we expected filaments to emerge from the protoplast, since we had never observed yeast-cells for this fungus. Furthermore, the slow growth rate of *T. thlaspeos* cultures suggests a longer regeneration time. To confirm our expectation, we described the regeneration process and its timing for *T. thlaspeos* protoplasts on regeneration medium with 1 M sucrose. After one day, protoplasts turned dark, which is indicative of cell wall regeneration. Three to eight days later, a filament emerges from the protoplast that starts branching after 7–13 days, finally resulting in a micro-colony after 11–18 days (Figure 4C). Further proliferation leads to filamentous colonies after four to five weeks, which are indistinguishable from the original culture.

### 3.2. Transformation

Five antibiotic resistance markers directed against phleomycin, hygromycin, nourseothricin, geneticin, and carboxin are routinely used in *U. maydis* [3]. To develop markers for *T. thlaspeos*, we tested culture growth on four of these antibiotics. Phleomycin is a mutagen and therefore was not considered [42]. *T. thlaspeos* cells were efficiently killed by the four antibiotics (Appendix A). Concentration gradients with hygromycin, nourseothricin, and carboxin revealed that *T. thlaspeos* was more sensitive towards these antibiotics than *U. maydis*. 10 µg/ hygromycin mL and 50 µg/mL nourseothricin efficiently killed *T. thlaspeos* hyphae. This is 20 times and three time less than the standard concentration used for *U. maydis,* respectively. By contrast, cells are less sensitive towards carboxin and remained resistant at 2 µg/mL, the standard concentration used for *U. maydis*, but were sensitive at 100 µg/mL. (Appendix A). Carboxin inhibits the mitochondrial succinate dehydrogenase (SDH2), and a point mutation, H253L, leads to a resistant form in *U. maydis* [43]. The *T. thlaspeos* SDH2 was highly conserved, with 82% amino acid similarity, and contained an arginine instead of the histidine at this position (Appendix A). This might explain the reduced sensitivity.

Due to the high hygromycin sensitivity, the bacterial hygromycin-phospho-transferase (*hpt*) [17] was used as the first resistance marker. We expressed it as a fusion protein hpt–eGfp [44] under the control of *T. thlaspeos* and *U. maydis* promoters and terminators (Appendix A). Promoter activity was verified in *U. maydis*. Both P*_Tt_*_hsp70_ and P*_Tt_*_rps27_ were active in *U. maydis*, and all five constructs resulted in hygromycin-resistant transformants with eGfp-fluorescence (Figure 5 and Appendix A), suggesting the constructs are functional and can be used to transform *T. thlaspeos*.

First, we needed to define which plasmid to use. Therefore, we transformed an equimolar mixture of five hpt-eGfp plasmids with different promoters (Appendix A) into *T. thlaspeos* LF1 protoplasts generated with the optimized method using the standard *U. maydis* conditions for transformation [28]. This resulted in a single transformant which had stably integrated the P*_Tthsp70_*::*hpt-egfp*:T*_Tthsp70_* into the genome (Appendix A, Figure 6). Now, transformations using this plasmid regularly result in fluorescent transformants. Based on this successful transformation, a P*_Tthsp70_*::*hpt*-*mcherry*:T*_Tthsp70_* construct was generated (Appendix A), tested in *U. maydis* (Figure 5), and transformed into *T. thlaspeos* LF2 (Figure 6), showing that compatible mating partners of *T. thlaspeos* can be tagged with different reporters to follow the mating process.

One key aspect for gene targeted manipulations is efficient homologous recombination. To test whether *T. thlaspeos* reaches the same high rates of up to 50% as *U. maydis*, we targeted the pheromone receptor gene *pra1* in the *T. thlaspeos* LF1 background for deletion. The construct design was based on *U. maydis* with 1 kb flanking sequences [27]. Transformation of the construct resulted in 122 candidates on the transformation plates. Reselection of 19 candidates on fresh hygromycin plates led to only nine candidates that remained resistant. The other candidates were either false positives, or they only transiently expressed the resistance protein. These are not interesting for stable integration. In subsequent analysis of the nine candidates, successful deletion of the *pra1* locus was confirmed for two transformants (Appendix A), giving a homologous recombination rate of 22%.

In summary, we have now adapted the protoplast-mediated transformation for the filamentously growing Brassicaceae smut fungus *T. thlaspeos*. Together, with its ability for efficient homologous recombination, this gives us a tool to study plant–microbe interactions of smut fungi in the model plant *A. thaliana* with two genetically tractable partners.

### 3.3. Mating of Filaments

When *T. thlaspeos* teliospores germinate, they give rise to an infectious filament that can directly penetrate the plant. On the other hand, these filaments also can give rise to haploid culture. Our haploid cultures *T. thlaspeos* LF1 and LF2 have compatible mating types. They can fuse at the tip and form a new filament [15]. To visualize directional growth of compatible LF1 and LF2 hyphae towards each other during mating, we carried out confrontation experiments. In close proximity, LF1 and LF2 hyphae sense each other, and reorient their growth to meet (Figure 7A). In some cases, some hyphae return their growth in direction towards the compatible filament after initial passage. Upon contact, they fuse and result in a new filament (Figure 7A,B, Appendix A). To prove that fused hyphae really share a common cytoplasm, mating was also observed in cocultivation experiments of compatible strains expressing eGfp and mCherry (Figure 7C). After hyphal fusion, eGfp and mCherry fluorescence could be observed in one cytoplasmic segment indicative of plasmogamy. On the other hand, if the pheromone receptor Pra1 is deleted, hyphae of compatible strains grow directly past each other without hyphal fusion (Figure 7A,B and Appendix A). These findings confirm that the pheromone-receptor system in *T. thlaspeos* [15] is active and initiates mating. In the future, the generation of nuclei-reporter-strains with NLS-fusion-constructs will allow tracking of the nuclei and thereby the investigation of karyogamy during mating.

## 4. Discussion

When we first set out to work with *T. thlaspeos*, our aim was to establish a genetically tractable smut fungus in a model host plant such as *A. thaliana* [15]. An important aim for reaching this goal was genetic manipulation. Here, we show that like other smut fungi, *T. thlaspeos* is amenable to protoplast-mediated transformation. We have generated a hygromycin resistance cassette, where expression of the hygromycin-phospho-transferase, hpt, is controlled by the *T. thlaspeos hsp70* promoter sequence, similar to the cassettes used in *U. maydis* [3]. Interestingly, promoter sequences seem to be exchangeable between smut fungi, since the *T. thlaspeos* promoters were active in *U. maydis* and several groups have successfully used *U. maydis* constructs [23,45]. This now enables us to generate reporter strains for a broad range of scientific questions.

Most important to establishing a successful protoplasting protocol is the choice of the lytic enzyme(s). The fungal cell wall is a multilayered, chemically complex structure consisting mainly of polysaccharides and varying amounts of lipids, proteins, and polyphosphates [46]. Its composition is not only variable between species [30], but also highly dependent on the culture conditions [47] and morphology [48]. In our case, the combined activity of Yatalase and Glucanex was necessary for efficient digestion of the *T. thlaspeos* cell wall, although they appear to have overlapping enzymatic properties. Similar additive effects have recently been shown for the ascomycete *Hirsutella sinensis* [49] and *Ag. aegerita* [31]; while in *Cordyceps militaris,* the enzymes mix is less active than Glucanex alone [50]. Hence, during the establishment of conditions for protoplasting, various lytic enzymes and combinations should be tested to reach optimal cell wall degradation [30]. Moreover, commercial manufacturing of enzymes can be stopped, with the broadly used Novozyme 234 being a recent example. Hence, the identification of suitable enzymes can be a reoccurring problem even for established protocols.

The second important factor is the osmotic stabilizer, because it depends on the choice of the protoplasting enzyme. For example, the enzymatic activity of Yatalase is inhibited by sorbitol and sucrose. In the 1970s, similar observations were made for helicase [35] and snail enzyme [34]. Protoplasting protocols with Yatalase use inorganic salts as osmotic stabilizer [21,31,37,40,51] and similar to these old reports, for *T. thlaspeos*, we now use MgSO_4_ to enable cell wall degradation.

Together with other factors influencing the protoplast formation, such as growth conditions of the culture, buffer composition, pH, temperature, or protoplasting time, establishing new transformation protocols quickly becomes a multi-factorial challenge, and testing full-factorial replicates is time-consuming and costly. For *T. thlaspeos*, we designed pilot studies covering selected combinations in single replicates based on existing transformation protocols, and used the most promising buffer, osmotic stabilizer, and enzyme combination for further optimization. While this approach does not cover all combinations, it allowed us to establish a good transformation protocol with reasonable effort.

As a proof-of-principle for our transformation protocol, we investigated the well-characterized smut fungal mating process in *T. thlaspeos*. In the first step, we generated reporter strains expressing cytosolic eGfp or mCherry to visualize the fusion of hyphae. The resulting filaments express both eGfp and mCherry, indicative of a common cytoplasm, as typical for the dikaryotic smut fungi [52,53]. Next, we looked into dependency on the pheromone receptor. To this end, we generated a deletion mutant of the pheromone receptor *pra1* [15] based on the strategy of *U. maydis* [28]. Notably, homologous recombination also takes place in *T. thlaspeos*, so we can modify genes in the haploid culture background easily.

*T. thlaspeos pra1* deletion mutants cannot mate anymore. This finding is especially interesting since it is not yet known whether mating is required for *T. thlaspeos* to fulfil its life cycle. Infectious filaments emerge directly from germinating *T. thlaspeos* teliospores. By contrast, teliospore germination of grass smut fungi gives rise to yeast-like sporidia. Subsequent pathogenic development depends on the morphological switch from yeast to filamentous growth brought about by mating [52]. However, the functional conservation of mating genes in *T. thlaspeos* suggests an evolutionary-conserved, and therefore important, role of mating also in this fungus [15]. This raises several questions. Is mating necessary for the lifecycle of *T. thlaspeos*? Where and when does mating occur? When do *T. thlaspeos* hyphae undergo meiosis? Is the filament emerging from the teliospore diploid or dikaryotic? Is the transition to haploid hyphae also occurring naturally in this state of the lifecycle? With the established transformation protocol, we will be able to further address these questions. This will shed light not only onto the mating process of *T. thlaspeos*, but also on the role of RNA communication in virulence, perennial persistence of the fungus *in planta*, and nutrition of a smut fungus during biotrophic growth.

## 5. Conclusions

Establishing the genetic manipulation of the Brassicaceae smut fungus *T. thlaspeos* now allows us to generate reporter strains as well as targeted deletions or modifications of fungal genes. Combined with the fungal colonization of the model plant *A. thaliana*, we thereby provide a pathosystem, in which both partners have a small, genetically tractable genome for addressing the current and future questions of plant–microbe interactions.

## Figures and Tables

**Figure 1 jof-07-00038-f001:**
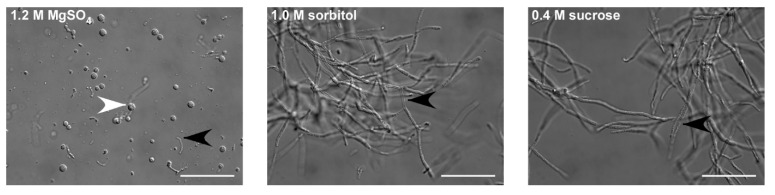
Identification of an osmotic stabilizer. *Thecaphora thlaspeos* LF1 culture was grown to an OD_600_ = 0.4–0.8. To optimize protoplasting of *T. thlaspeos* hyphae by Yatalase and Glucanex, the osmotic stabilizers MgSO_4_, sorbitol, and sucrose were tested. With MgSO_4_ as osmotic stabilizer, all filaments were digested; while in sorbitol and sucrose, no protoplasts were obtained. Black arrowheads: filaments; white arrowheads: protoplast; scale bar: 50 µm.

**Figure 2 jof-07-00038-f002:**
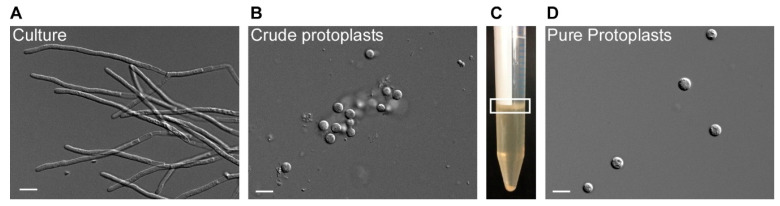
Protoplasting filamentous *T. thlaspeos* cultures. 1–2 g fresh weight of an exponentially growing culture (**A**) was harvested by filtration and protoplasted with Glucanex and Yatalase for 30–60 min. To purify the crude protoplasts (**B**), they are floated in a gradient (**C**). Filaments and debris are found in the pellet, the protoplasts can be collected from the interphase (marked with a white box). (**D**) An efficient protoplasting reaction results in up to 10^8^ protoplasts/g fresh weight. Scale bar: 10 µm.

**Figure 3 jof-07-00038-f003:**
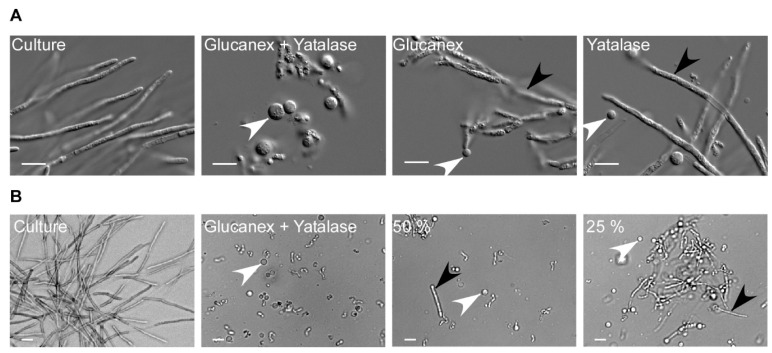
Optimizing the enzyme cocktail for protoplasting. (**A**) Filamentous cultures were harvested and incubated with a combination of 20 mg/mL Glucanex and 10 mg/mL Yatalase, or each enzyme individually, for 30 min. Protoplasting was efficient only when both enzymes were applied. (**B**) The enzymes were diluted to identify the lowest suitable concentration. The frequency of remaining filaments is inversely proportional to the enzyme concentration. The highest efficiency was obtained with 20 mg/mL Glucanex and 10 mg/mL Yatalase. White arrowhead: protoplasts; black arrowhead: residual filaments; scale bar: 10 µm.

**Figure 4 jof-07-00038-f004:**
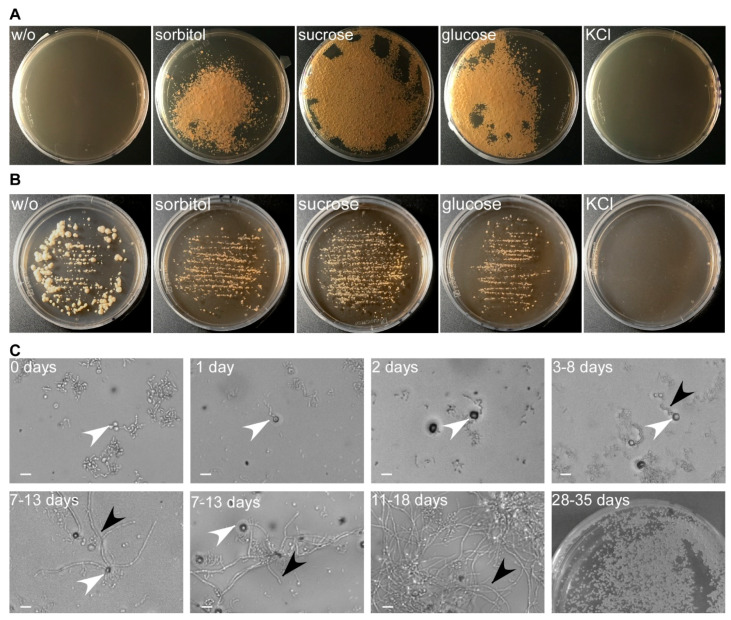
Optimizing protoplast regeneration. (**A**) Protoplasts regenerated for 39 days on regeneration medium (YMPG) with either 1 M sorbitol, sucrose, glucose, or KCl. No regeneration was observed without osmotic stabilizer. It was most efficient on sucrose, followed by glucose and sorbitol, while KCl inhibited cell growth. (**B**) *T. thlaspeos* LF1 culture plated on YMPG with sorbitol, sucrose, glucose, KCl, or without osmotic stabilizer. After 21 days, no growth was observed for 1 M KCl. Growth rate is reduced for 1 M sorbitol, glucose, and sucrose compared to absence of osmotic stabilizer. (**C**) Regeneration of protoplasts on YMPG with 1 M sucrose documented microscopically for 11–18 days. Initially, the protoplasts turned dark (2 days) before new filaments emerged (3–8 days). The filaments proliferated (11–18 days), resulting in macroscopically visible colonies (28–35 days). White arrowhead: protoplasts; black arrowhead: emerging filaments; scale bar: 10 µm.

**Figure 5 jof-07-00038-f005:**
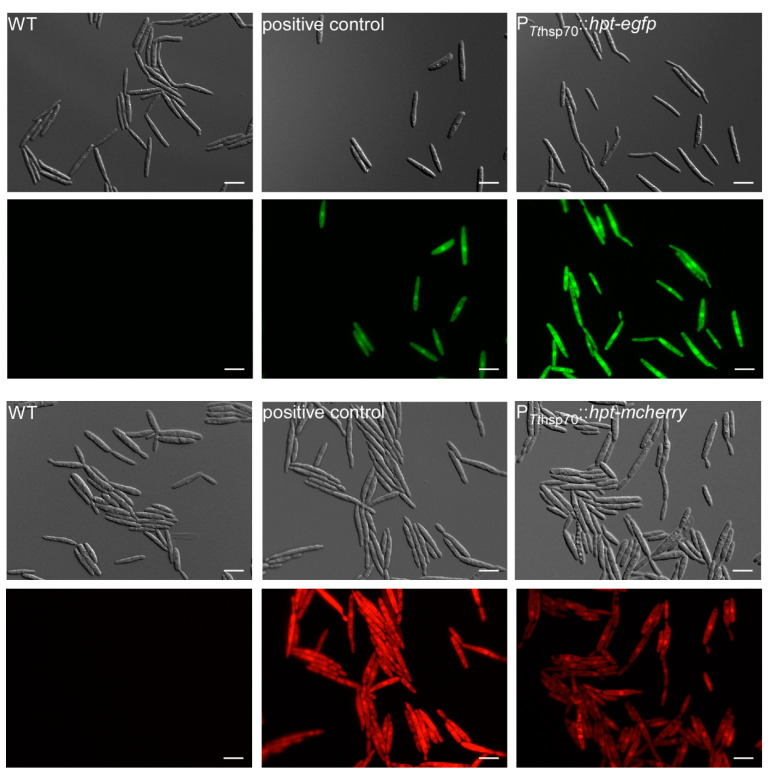
Verification of resistance-reporter constructs in *U. maydis*. Reporter constructs containing a fusion of hygromycin-phospho-transferase gene (*hpt*) and the fluorescent marker (*egfp* or *mcherry*) under the control of *hsp70* promoter and terminator regions derived from the *T. thlaspeos* genome were tested in *U. maydis*. Upon transformation of the linearized construct, it randomly integrates into the genome. Protein accumulation was visualized by the green/red fluorescence. The eGfp expression under the promoter region of *T. thlaspeos* was stronger than compared to the stably integrated construct under the control of a strong, synthetic promoter (P*otef*). This confirms that the fusion protein is active. In comparison, mcherry-fluorescence in the strain carrying the *Tthsp70* promoter was weaker than the stably integrated construct under the control of the P*otef* promoter. Scale bar: 10 µm.

**Figure 6 jof-07-00038-f006:**
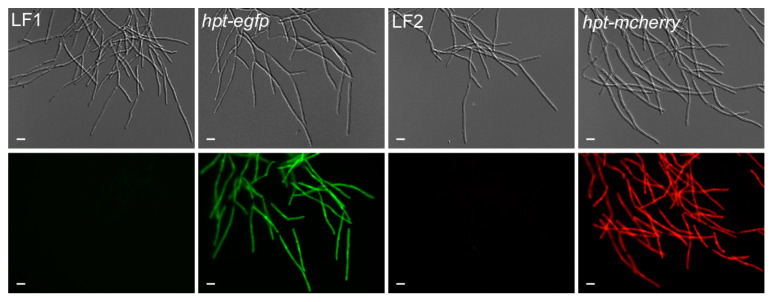
Generation of reporter lines in *T. thlaspeos*. Reporter constructs containing the active fusion of the hygromycin-phospho-transferase gene (*hpt*) and the fluorescent marker (*egfp* or *mcherry*) under the control of the strong hsp70 promoter from *T. thlaspeos* were transformed into the cultures *T. thlaspeos* LF1 or LF2, respectively. Fluorescent signals accumulate in the cytosol of all cells. Strains: hpt-egfp: LF1_P*_Tthsp70_*::*hpt-egfp*, and hpt-mcherry: LF2_P*_Tthsp70_*::*hpt-mcherry*. Scale bar: 10 µm.

**Figure 7 jof-07-00038-f007:**
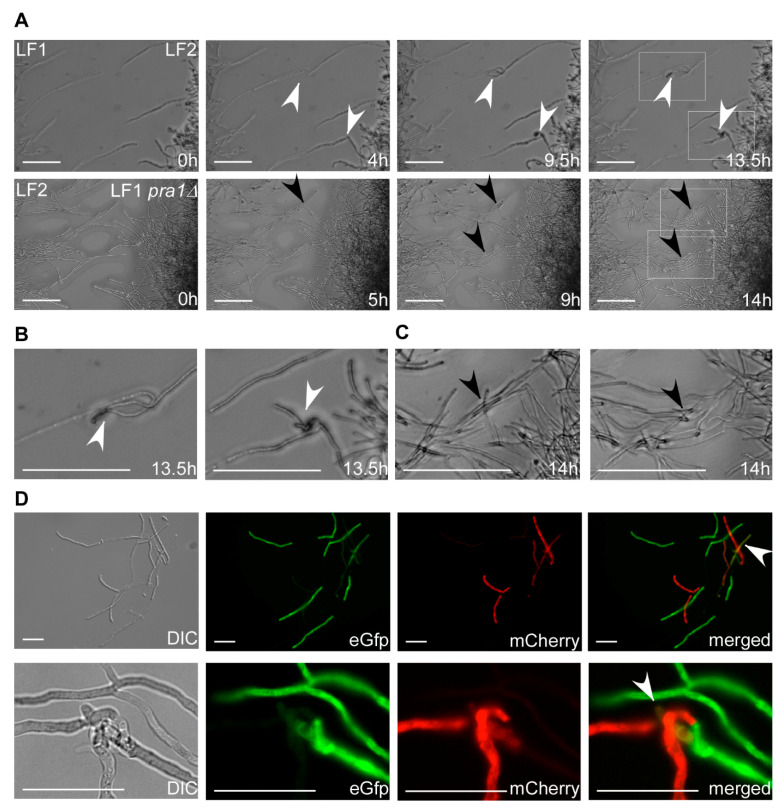
Mating in *T. thlaspeos.* (**A**) Top: Confrontation assay of mating partners *mfa1/pra1* (LF1) or *mfa2/pra2* (LF2). Over the course of 13.5 hours, several mating events of LF1 and LF2 were observed. White arrowheads mark the fusion event. Bottom: Confrontation assay of LF1 *pra1∆* and LF2. Over the course of 14 h, hyphae did not mate. Two spots where hyphae could no longer sense each other and cross are marked with black arrow heads. Scale bar: 25 µm. (**B**,**C**) Magnification of the mating and crossing events from the white boxes marked in (**A**). Scale bar: 100 µm. (**D**) Liquid mating assay with mating partners expressing either cytoplasmic eGfp (LF1-eGfp) or mCherry (LF2-mCherry). Fused hyphae express both eGfp and mCherry and appear yellow in the merged picture (white arrowhead). Scale bar: 25 µm.

**Table 1 jof-07-00038-t001:** Optimizing the protoplasting buffer. To identify the optimal osmotic stabilizer, fungal hyphae were filtered and incubated in 0.02 M citrate buffer, supplemented with different osmotic stabilizers and 10 mg/mL Yatalase + 20 mg/mL Glucanex, for 60 min at RT. Protoplasting worked only if MgSO_4_ was used as osmotic stabilizer. To optimize the buffer for the use of MgSO_4_, hyphae were filtered and incubated in different buffers, supplemented with 1.2 M MgSO_4_ and 10 mg/mL Yatalase + 20 mg/mL Glucanex, for 60 min at RT. There was no significant difference between the indicated buffers, but a tendency towards higher yields with citrate buffers.

Buffer	Osmotic Stabilizer	Protoplast Yield/g FW × 10^7^
Optimizing the Osmotic Stabilizer		
0.02 M citrate, pH 5.8 [28]	0.4 M sucrose [20]	no protoplasts
	1.2 M MgSO_4_ [36]	5.52 ± 1.22
	1.0 M sorbitol [28]	no protoplasts
Optimizing the buffer composition		
0.1 M citrate, 0.01 M EDTA, pH 5.8 [23]	1.2 M MgSO_4_ [36]	7.46 ± 2.02
0.02 M citrate, pH 5.8 [28]		7.49 ± 1.51
0.02 M MES, pH 5.8 [21]		5.28 ± 0.66
0.01 M phosphate, pH 5.8 [37]		5.01 ± 0.51

## Data Availability

The data presented in this study are available in this manuscript and constructs can be requested from the corresponding author.

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
