# Peer review of "Genetic Manipulation of the Brassicaceae Smut Fungus *Thecaphora thlaspeos"

_jof, 2021, doi:10.3390/jof7010038_

Round 1

Reviewer 1 Report

This manuscript by the Göhre lab reports on a genetic system for the Brassicaceae smut fungus Thecaphora thlaspeos. As far as this reviewer can judge, English language in this manuscript is fine apart from a very few small errors. All in all, this reviewer has very much enjoyed reading this manuscript as it only makes one major comment necessary:

Although the authors described that they did it in the ‘Material and Methods’ section, the authors somehow missed to show their PCR- and Southern analysis-based proofs of transformation construct integration into as well as pra1 deletion from the fungus’ genome.

I made all my comments, including the mentioned major comment, directly in the submission pdf file. This reviewer thinks that, apart from the major shortcoming (was probably just forgotten to include in the submission?), this study otherwise has been carried out very thoroughly.

One last comment about the ratings placed with this manuscript by the reviewer:

- This reviewer, sees the merits of this report from a rather (not exclusively!) “basidiomycete-based perspective”. Therefore, this reviewer finds it tenable to judge the merits of this study (novelty, significance etc.) as “high” to (basidiomycete) fungal biology and genetics. The only subtraction regarding the “scientific soundness” is not placed based on the overall thorough experimental design and realization: downrating is just based on the missing the PCR- and Southern analysis-based proofs of transformation construct integration into as well as pra1 deletion from the fungus’ genome that are not shown and need to be added before recommendation for publication can be considered.

- Also the “must be improved”-judgment on the presentation of the results solely refers to the missing presentation of Southern analysis and PCR data. The rest of the data is clearly presented.

All in all, this reviewer, who would otherwise opt for a minor revision, has to opt for a major revision because of the missing PCR and Southern analysis data before the manuscript can be recommended for publication.

Author Response

Thank you for your very detailed feedback! We are happy to include our molecular analysis of the strains (Southern Blot and PCR analysis). They are now added as supplementary figures and described in the text (see also below). We have also addressed the minor points commented in the pdf as detailed here:

  • Manufactures are added
  • Information on cryo-conservation of T. thlaspeos is added in material and methods
  • Details on the Golden Gate cloning and the plasmid maps are now added in protocol S1
  • The selection in 2-layered plates is now better explained in the text. We also tried to supplement the media with hygromycin after different regeneration times as suggested, but this was less practical and did not improve the transformation protocol in our hand. However, in the future, this might be an interesting alternative to test to initially limit the number of false positives prior to re-streaking.
  • For the reporter lines, we have added the PCR-based confirmation of the integration in Figure S5.
  • For the pra1 mutant, we have added the PCR-confirmation and the Southern blot in figure S5.
  • We have added the citation Herzog 2019 for the use of the combination of Yatalase and Glucanex in Agrocybe both in the results and the discussion.
  • We did not consider malate buffers, since we obtained good protoplasting reactions with the citrate buffer. However, it will be interesting in the future to also test malate buffers for protoplasting to see if this can improve regeneration.
  • Reducing the background of false positives was not a major concern on our case, since it is relatively easy to re-streak many candidates for selection. In the future, as detailed above, we will for sure include modified regeneration protocols.
  • “One might want to move this down to Discussion as these are hypotheses, but this reviewer thinks that this may not be absolutely necessary…” – we decided to shorten the sentence, but leave it in the result sections, since it gives some details on the false positives, which always occur during transformations. In the discussion, we do not get back to this minor point, and moving this sentence would lengthen the discussion.
  • Conclusion: we have re-phrased the sentence to tune down the statement on all genes can be modified, which is clearly a generalisation.

Reviewer 2 Report

The manuscript describes a new method for the genetic manipulation of the important plant pathogen Thecaphora thlaspeos. The paper is well written, the methodology is clear and adequate, the conclusions logical. All the important parameter for the new technology were tested and optimized, the figures are clear and informative. A few minor errors can be easily fixed: e.g. line 49 (germinating), line 263 (First, we first…). Species names should be written in italics throughout the text.

Author Response

Thank you for your very positive feedback! We have corrected the errors and now made sure that the species names are in italics. This was a file conversion problem that escaped our attention. Sorry for that!

Round 2

Reviewer 1 Report

This reviewer very much appreciates the amendments the authors have implemented to the manuscript and supplement. There are only a few small issues left where the authors may want to apply cosmetic changes before the paper should be published.

I made all my comments in the respective submission pdf files.

This reviewer thinks that the authors may want to once more polish the manuscript by the recommended small optimisations (and, thus, ticked the check box “can be improved” where it is asked “Are the results clearly presented”?) and opts for a very minor revision.

Author Response

Thanks for pointing out this minor mistakes and the suggestion to add a schematic overview of the Southern Blot.

We now refer to Figure S5 in the text twice, when we explain the reporter strains and the deletion mutant. 

The legend of protocol S1 is corrected. We added more information on IDT and corrected the formatting errror.

In figure S5, we added a scheme of the genetic loci as suggested as part (D) of the figure.

The formating of restriction enzymes has changed, and we no longer use the itacicised version. There are no strict formatting instructions in the guidelines regarding enzymes.